# A Comparative Analysis of Fasciotomy Results in Children and Adults Affected by Crush-Induced Acute Kidney Injury following the Kahramanmaraş Earthquakes

**DOI:** 10.3390/medicina59091593

**Published:** 2023-09-03

**Authors:** Mustafa Yalın, Fatih Gölgelioğlu

**Affiliations:** Department of Orthopaedics and Traumatology, Elazığ Fethi Sekin City Hospital, Elazığ 23050, Turkey; fatihgolgelioglu@gmail.com

**Keywords:** crush-related acute kidney injury, fasciotomy, morbidity, earthquake

## Abstract

*Background and Objectives*: The current study aims to determine the impact of fasciotomy on mortality and morbidity in children and adults with crush-related AKI following the 2023 Kahramanmaraş earthquakes. *Materials and Methods*: The study included individuals who had suffered crush injuries after the 2023 Kahramanmaraş earthquakes and were identified as having an acute kidney injury (AKI). Patients with an AKI were divided into two groups based on age: those under 18 years and those over 18 years. A comparative analysis was conducted between the mortality and morbidity rates of patients who underwent fasciotomy and those who did not. Disseminated intravascular coagulopathy (DIC), sepsis, and adult respiratory distress syndrome (ARDS) have all been identified as contributors to morbidity. *Results*: The study was conducted with a total of 40 patients (21 males and 19 females) aged between 4 and 83 years. A total of 21 patients underwent fasciotomy, and the patients underwent varying numbers of fasciotomy, ranging from 0 to 11. The mortality rate was 12.5%, corresponding to five adult patients. No instances of mortality were reported in the paediatric cohort. The application of fasciotomy in instances of crush-induced AKI did not result in elevated levels of mortality in either the paediatric or adult demographic. Within the adult population, a substantial difference in the duration of dialysis was observed between individuals who underwent fasciotomy and those who did not. A statistically significant increase in the number of fasciotomy incisions was observed in patients diagnosed with sepsis compared with those without sepsis. The study found a significant positive correlation between the number of fasciotomy incisions and dialysis days. *Conclusions*: Neither adult nor paediatric patients with crush-induced AKI showed an increased risk of death after fasciotomy. The number of fasciotomy incisions significantly correlated with the development of sepsis. Despite experiencing delays in hospital admission for paediatric patients, the incidence of both crush syndrome and mortality rates among children remained relatively low.

## 1. Introduction

On 6 February 2023, Kahramanmaraş, southeastern Turkey, experienced two devastating earthquakes. The first earthquake had a magnitude of 7.7 on the Richter scale, followed by a second earthquake nine hours later, with a magnitude of 7.6. These back-to-back earthquakes were accompanied by over 1000 aftershocks, some of which exceeded a magnitude of 6. The impact of these consecutive earthquakes has been unprecedented in recent Turkish history, causing widespread destruction and resulting in a current estimate of 35,000 reported fatalities and 105,000 injured individuals within one week of the earthquakes [1]. In the aftermath of the earthquake, the widespread destruction of medical facilities posed a significant challenge to delivering immediate healthcare to the affected population. Consequently, a considerable number of patients must be transferred to hospitals located on the periphery of the affected area. Notably, the emergency department (ED) of our institution has emerged as a vital hub for receiving a substantial influx of patients from neighbouring provinces. This strategic measure played a pivotal role in mitigating the burden on the strained local health system, thereby enabling the hospital to allocate approximately 1000 beds exclusively to address the urgent needs of earthquake victims. 

Earthquake catastrophes not only lead to a significant number of instantaneous fatalities due to injury to essential organs, but also bring with them a cluster of severely injured victims, in whom crush incidents and prolonged entrapment of extremities are prevalent types of trauma. When a limb is trapped under debris, muscle cells experience mechanical strain, leading to the release of their cellular contents [2]. Increased intracompartmental pressure may result from an increase in both interstitial and intracellular fluid volumes. When the pressure surpasses the capillary perfusion pressure, which is approximately 30 mmHg, muscular veins collapse. Furthermore, tissue oxygenation is disrupted if the pressure exceeds the diastolic blood pressure threshold. Consequently, the resulting ischemic damage causes necrosis in the muscles, known as rhabdomyolysis, as well as in nerves [3]. According to estimates, acute compartment syndrome may lead to muscle necrosis in up to 35% of patients within 2 h following the injury [4]. The consensus in the medical community is that prompt initiation of early fasciotomy is crucial to achieve the best outcomes in cases of compartment syndrome [5,6,7,8,9,10]. Disruption of renal blood flow due to hypovolemia and the accumulation of nephrotoxic deposits increase the likelihood of acute kidney injury (AKI) in patients with crush injury [11]. In this already problematic scenario, fasciotomy’s open incision might worsen preexisting coagulopathy and increase the risk of death from sepsis. Dialysis for patients with an AKI will be more difficult, and the risk of an AKI will increase as a result of these factors. The current study aimed to determine the impact of fasciotomy on mortality and morbidity in children and adults with crush-related AKIs following the Kahramanmaraş earthquakes in Turkey. 

## 2. Materials and Methods

Injuries from the 2023 Kahramanmaraş earthquakes were the focus of the current retrospective investigation conducted at a single centre. The present study was approved by the local ethics committee (Approval Number: 2023/07-12). In accordance with established ethical guidelines, written informed consent was obtained from all participants, thereby ensuring voluntary participation and authorisation for the utilisation of their anonymised data in the current study. The study strictly adhered to the principles outlined in the Declaration of Helsinki governing the ethical conduct of clinical research. Stringent measures were implemented throughout the study to maintain patient confidentiality and to safeguard the privacy of their personal information. The study included individuals who had suffered crush injuries subsequent to an earthquake and were identified by experts as having an AKI based on the analysis of biochemical markers. The information for this research was gathered through a comprehensive review of existing medical files. X34 (earthquake victim) identification was issued to individuals who self-reported injuries sustained during the earthquake or were transported via ambulance from a nearby province. Disseminated intravascular coagulopathy (DIC), sepsis, and adult respiratory distress syndrome (ARDS) have all been identified as contributors to morbidity. Patients with a prior medical condition of chronic renal failure or an AKI resulting from unrelated factors were excluded from the study. Additionally, individuals with injuries or circumstances that could not be directly attributed to earthquakes were excluded. Patients diagnosed with an AKI were stratified into two distinct age cohorts: those below and those above the age of 18. A comparative analysis was conducted within each age group and between the two age groups. The objective of the current study was to conduct a comparative analysis of the demographic characteristics, biochemistry profiles, and clinical outcomes of patients who were diagnosed with an AKI, with a specific focus on whether they underwent fasciotomy. A comparative analysis was conducted on the mortality and morbidity rates of patients who underwent fasciotomies and those who did not. The decision of whether or not to proceed with a fasciotomy operation was made by in-clinic evaluations conducted by physicians with a minimum of five years of expertise in trauma medicine. These assessments included both the viability of the affected limb and the cost–benefit analysis associated with performing the fasciotomy. Patients who received emergency fasciotomy upon initial presentation, as well as those who were eligible for fasciotomy but needed careful monitoring, were admitted to the intensive care unit. The patients were closely monitored by nephrology and intensive care physicians, with regular visits from orthopaedic doctors to assess their circulation and wound healing. The fasciotomy procedures involved the performing of two wide incisions, one on the medial side and the other on the lateral side, for the cruris and thigh. Additionally, two incisions were made in the dorsal region of the foot, specifically in the second and fourth web intervals. A wide volar incision was performed on the forearm, while three incisions were made in the dorsal region of the hand, specifically in the second and fourth web intervals and the tenar region. In the current study, all individuals who had fasciotomy were subjected to vacuum-assisted debridement and irrigation at regular 3-day intervals. Additionally, the wound lip approximation method, aimed at facilitating quicker closure of the wound lips, was performed during each debridement process. In cases where primary delayed closure was not achievable due to skin retraction, full-thickness skin grafting was scheduled for all patients. Support for this procedure was sought from the plastic surgery department. Wound culture samples were routinely taken from patients during the debridement process to investigate suspected infections at the wound site or any deterioration observed during clinical follow-up. The results of these samples were then referred to the infectious diseases department for consultation. In individuals diagnosed with foot drop or radial nerve palsy, the use of splints and orthoses was promptly initiated in order to maintain joint neutrality and stability. Based on the outcomes of in-clinic assessments, the decision was made to proceed with amputation of the limb that experienced a loss of viability.

### 2.1. Acute Kidney Injury Definition

Owing to the retrospective nature of the study, conventional AKI criteria were not used [12]. In the case of crush-related acute kidney injury, the victim must have had a crush injury and have shown at least one of the following symptoms: serum potassium > 6 mEq/L, phosphorus > 8 mg/dL, or calcium < 8 mg/dL; oliguria (urine production < 400 mL/day); blood urea nitrogen > 40 mg/dL; serum creatinine > 2 mg/dL; and serum uric acid > 8 mg/dL [13].

### 2.2. Definitions of Disseminated Intravascular Coagulation (DIC), Acute Respiratory Distress Syndrome (ARDS), and Sepsis

The diagnosis of DIC was determined by integrating clinical observations with laboratory measurements [14]. The definitive diagnosis of DIC can only be established when there is a recognised primary ailment that has been determined to be linked with DIC and when the clinical manifestations and symptoms align with this underlying medical condition. Common results in routine laboratory tests include thrombocytopenia as well as a rapid decline in the number of platelets; unusual monitoring assessments such as prothrombin time (PT) or activated partial thromboplastin time (aPTT); and a significant increase in markers of the formation of fibrin and subsequent collapse, such as D-dimer or other forms of fibrin products of degradation [14]. Thromboembolism affecting larger blood vessels will manifest with symptoms consistent with the blockage of blood flow. Another common observation is the occurrence of extensive bleeding from mucosal tissue, such as the gingiva, nose, or digestive tract, as well as from the insertion sites of indwelling catheters [15]. Patients suspected of having DIC were referred for haematology consultations as part of their intensive care unit (ICU) follow-up. The final diagnoses were established in collaboration with the haematology department.

ARDS is characterised by sudden hypoxemia and bilateral pulmonary oedema caused by increased permeability of the alveolocapillary membrane. ARDS was defined clinically by the Berlin description (panel 1) and included stages that assess mortality risk. However, there is no singular test available to definitively diagnose or rule out ARDS [16]. The pulmonology department diagnosed and monitored ARDS during the ICU follow-up.

Sepsis is characterised as a severe condition in which the body’s response to infection becomes uncontrolled, leading to life-threatening dysfunction of organs. The clinical criteria for sepsis consisted of suspected or confirmed infection and an unexpected rise of a minimum of two Sequential Organ Failure Assessment (SOFA) points, which served as an indicator of failure of the organs [17]. The diagnosis of sepsis was established by the analysis of culture samples obtained during vacuum-assisted debridements, in conjunction with an assessment of the patients’ clinical condition, in collaboration with the infectious diseases department.

### 2.3. Statistics

Statistical analyses of the study’s findings were conducted using IBM SPSS Statistics 22 software (Version 22.0. Armonk, NY, USA: IBM Corp; 2013; IBM SPSS, Elazığ, Turkey). The normality of the parameters was assessed using the Shapiro–Wilk test. The study employed both descriptive statistical techniques, such as means, standard deviations, and frequencies, as well as inferential statistical methods. Specifically, the Student t test was used to compare quantitative data between two groups with normally distributed parameters, while the Mann–Whitney U test was employed to compare non-normally distributed parameters between the two groups. Fisher’s exact test was used to compare the qualitative datasets. Spearman’s rho correlation analysis was employed to examine the associations among variables that did not adhere to a normal distribution. Statistical significance was set at a significance level of *p* < 0.05. 

## 3. Results

The current study was conducted with a total of 40 patients: 21 (52.5%) males and 19 (47.5%) females, aged between four and 83 years, with a mean age of 32.63 ± 18.99 years. The study included a total of 40 participants, with 10 being paediatric patients and 30 being adult patients. The children’s ages ranged from 4 to 17 years, with a mean age of 10.6 ± 4.48 years. The age range of the adult participants in the study was 19 to 83 years, with a mean age of 39.97 ± 15.98. Table 1 represents the descriptive characteristics of the participants. A total of 21 patients (15 adults and 6 children) underwent fasciotomy, and the patients underwent varying numbers of fasciotomy, ranging from 0 to 11, with a mean of 2.43 ± 3.16. All paediatric patients who underwent a fasciotomy procedure were diagnosed with compartment syndrome upon arrival at the ED, prompting the immediate initiation of an emergency fasciotomy procedure. All fasciotomy operations conducted on paediatric patients included cruris and thigh fasciotomy procedures for the lower extremities. Fasciotomy procedures were performed on three adult patients during their ICU follow-up, after discussions held within the clinic. Emergency fasciotomy was performed in 12 adult patients who presented to the ED. Additional fasciotomies were subsequently performed in 5 patients due to the development of compartment syndrome in other extremities during follow-up. We admitted one patient for further treatment with below-knee amputation. Additionally, we performed pinky toe amputation in two patients due to the development of necrosis during follow-up. The mortality rate was 12.5%, corresponding to five adult patients. All deaths occurred within the initial week of ICU follow-up. The causes of death were cardiac arrest in three patients, cardiac arrest and DIC in one patient, and sepsis and DIC in one patient.

The study was conducted in two groups: 10 (25%) children and 30 (75%) adults. In the paediatric population, hospital admission time varied from 18 to 144 h, with a mean of 65.1 ± 36.18 and a median of 60. The hospital admission time for adults varied from 14 to 70 h, with a mean of 37.63 ± 17.29 and a median of 32. The time to admission from an earthquake (TAE) was notably greater in the paediatric population than in adults (*p* = 0.011, *p* < 0.05). However, no instances of mortality have been reported in the paediatric cohort. No statistically significant difference was observed between the groups in terms of the time under debris (TUD) and number of fasciotomies. Additionally, the most frequently occurring nerve injury in both groups was peroneal nerve injury (Table 2). 

Upon comparing both groups in relation to morbidity, it was observed that the children displayed two instances of DIC and one instance of sepsis. However, adults displayed five instances of DIC, eight distances of ARDS and seven instances of sepsis, although statistical significance could not be determined owing to an inadequate sample size (Table 3). 

Analysis of both groups, specifically those who underwent fasciotomy and those who did not, showed no statistically significant difference between the patients who underwent fasciotomy and those who did not, with respect to the manifestation of morbidity criteria such as ARDS and DIC in both groups (Table 4). In the paediatric group, the number of dialysis days ranged from 0 to 10 with a mean of 3.83 ± 4.49, whereas none of the patients who did not undergo fasciotomy required dialysis. Within the adult cohort, the duration of dialysis in patients who had fasciotomy varied from 2 to 30 days, with an average of 9.93 ± 7.47 days. In contrast, non-fasciotomised patients experienced a dialysis duration ranging from 0 to 17 days, with a mean of 2.27 ± 4.61 days. In the adult demographic, there was a notable difference in the duration of dialysis between individuals who underwent fasciotomy and those who did not (*p* < 0.05) (Table 4). 

A significant correlation between an increase in the number of fasciotomy incisions and the development of sepsis was one of the most important findings of the current study. The number of fasciotomy incisions in sepsis patients ranged from 0 to 11, with a mean of 5 ± 4.07. In contrast, non-sepsis patients had a range of 0 to 8 incisions, with a mean of 1.78 ± 2.59. The number of fasciotomy incisions in patients with sepsis was found to be statistically significantly higher than that in those without sepsis (*p* = 0.028; *p* < 0.05) (Table 5). The study found a significant positive correlation between the number of fasciotomy incisions and dialysis days, with a correlation coefficient of 60.1% (*p* = 0.000; *p* < 0.05) **(**Figure 1). 

## 4. Discussion

The research findings unveiled a significant revelation indicating that the implementation of fasciotomy in cases of crush-induced AKI did not lead to increased rates of mortality in either the paediatric or adult population. Another significant finding was the strong correlation between the onset of sepsis and the number of fasciotomy incisions. An additional discovery from the investigation indicated that despite delayed hospital admissions among paediatric patients, the rates of mortality and morbidity were statistically comparable to those observed in adults. 

In a study conducted by Zhang et al. [18] in 2012, the mortality rate of victims with crush-related AKIs in the Wenchuan earthquake was 10.96%, which is in line with our rate of 12.5%. According to the report, 98 fasciotomy procedures were conducted in 68 patients throughout the entire series, accounting for 32.2% of the cases. A total of 91 limbs were subjected to amputation in a cohort of 72 patients, accounting for 34.1% of the sample. In the current study, 21 patients underwent 97 fasciotomy incisions, accounting for 52.5% of the study population. Our amputation rates were far lower than those in the aforementioned study, with only two patients requiring amputation of their fifth toe owing to the advancement of necrosis. 

Our findings are consistent with those of a study conducted in 2001 by Sever et al. [13], who found that the highest rates of crush syndrome were observed in patients aged 20–59 years and that both crush syndrome and death rates in children were fairly low. However, there is a lack of consensus in the literature regarding the prognosis of children. After the Kobe earthquake, it was observed that the age group of 30–39 years had the most favourable prognosis, while the rate of death among children was also relatively low [19]. In the wake of the earthquake in Guatemala, the number of fatalities below the age of 20 was relatively low [20]. Based on the aforementioned report, an inverse relationship was observed between age and mortality in children, except for infants, who potentially experienced a different pattern due to co-sleeping with their parents [20]. It is debatable whether children would not have the same chance of survival as adults in a confined location or whether their smaller bodies would provide them with an advantage. 

A study conducted by Safari et al. (2011) revealed that the implementation of fasciotomy did not yield a statistically significant effect on the morbidity or mortality rates among individuals with crush-induced AKIs subsequent to the Bam earthquake [21]. This finding was consistent with the results of the present study. Unfortunately, there is a lack of references in the literature regarding which individuals should undergo surgical treatment, when to perform the procedure, and how the procedure should be performed [22]. According to a study conducted by Greaves et al. [23], patients who underwent fasciotomy for crush syndrome face a significant risk of uncontrollable bleeding, sepsis, and wound infection. Hence, the foremost therapeutic approach involves the use of mannitol to reduce compartment pressure, while also highlighting the avoidance of surgical intervention. According to Kang et al. [24], survival rate may increase with prompt intensive care. After fasciotomy, only 13% of individuals with lower-extremity compartment syndrome and foot drop improved, according to Bradley’s study [25]. Matsuoka et al. [26] found no evidence that fasciotomy enhanced the results of crushed patients. The International Committee of the Red Cross (ICRC) and the Arbeitsgemeinschaft für Osteosynthesefragen (AO) Foundation’s Guidelines for the Management of Limb Injuries in Disasters and Wars recommend performing an urgent fasciotomy between 0 and 8 h after injury if there are clinical signs of compartment syndrome [27]. It has been claimed that the efficacy of fasciotomy eight to twenty-four hours after a disaster is debatable. The decision to perform fasciotomy should be made only after a thorough evaluation of limb viability. In the current study, the duration of time spent beneath the debris for nearly all patients who underwent fasciotomy ranged from 8 to 24 h. There was a male child aged four who was trapped beneath the rubble for a duration of 36 h, alongside a female patient aged 33 years who remained trapped for a period of 30 h. During the course of treatment, the paediatric patient experienced successful recovery through repetitive debridement and skin grafting (Figure 2). Conversely, the female patient developed ARDS and sepsis. However, the patient was ultimately discharged after receiving repetitive vacuum and muscle debridement treatments, as well as skin graft and muscle flap treatments during the follow-up period. 

One of the most important findings of the current study is that the risk of developing sepsis increases with the number of fasciotomy incisions, even if the presence of fasciotomy alone is not statistically significant in the development of sepsis. Patients who underwent fasciotomy had a much higher risk of developing sepsis, as reported in 2002 by Erek et al. [28]. Nonetheless, no statistically significant association with mortality was observed. The current study showed that individuals who underwent fasciotomy had a higher frequency of dialysis sessions. However, this factor did not exert any influence on mortality outcomes. In the context of the earthquake that occurred in Turkey in 1999, approximately 70% of patients diagnosed with compartment syndrome underwent routine fasciotomy. Subsequently, 81% of these patients developed sepsis as a result of wound infection [29]. The current study documented the occurrence of sepsis in six of 21 patients who underwent fasciotomy. Acinetobacter Baumannii was detected in wound cultures of four patients, and sepsis resulted in the death of one of these patients. The wound cultures of two patients yielded Staphylococcus aureus, while the wound culture of one patient yielded Pseudomonas aeruginosa. The decline in sepsis rates and the relatively low infection rates observed in patients undergoing fasciotomy, despite a high number of dialysis days, can be attributed to the insights gained from the instructive phrase “Do not perform fasciotomy on every patient!” which was acquired subsequent to the occurrence of the Marmara earthquake in 1999. 

### Limitations and Strengths

The present study was a retrospective examination of earthquake-affected patients, but the study was subject to a number of restrictions. First, the current study was characterised by its single-centre design, limited sample size, and limited follow-up. The number of participants and their distribution were insufficient to provide a precise depiction of the magnitude of the entire earthquake. Second, because of the retrospective design of the study, standard criteria for AKIs were not used. The diagnosis was confirmed through patient descriptions of crush injuries and the presence of metabolic abnormalities. Third, following the occurrence of the earthquake, the orthopaedic team engaged in rotational shifts within the ED. During this period, it was noted that certain surgeons preferred to perform fasciotomy, while others were more inclined towards providing close follow-up care. The presence of clinician variability in follow-up techniques constitutes an additional factor contributing to the lack of consistency. Fourth, there is a lack of cost-effectiveness analysis. The analysis of healthcare resource utilisation has the potential to provide valuable insights for surgical decision-making and disaster planning. The current study represents a unique contribution to the existing literature, as it examines the impact of fasciotomy on both mortality and morbidity in paediatric and adult patients with AKIs resulting from crush injuries. An additional significant aspect of the study lies in its strong focus on the positive correlation between the number of fasciotomy incisions and increased vulnerability to sepsis development. 

## 5. Conclusions

In conclusion, neither adult nor paediatric patients with crush-induced AKIs showed an increased risk of death after fasciotomy. In addition, the number of fasciotomy incisions was significantly correlated with the development of sepsis. Despite experiencing delays in hospital admission for paediatric patients, the incidence of both crush syndrome and mortality rates among children remained relatively low. It is crucial to conduct prospective studies with a larger number of participants to get more robust and reliable findings.

## Figures and Tables

**Figure 1 medicina-59-01593-f001:**
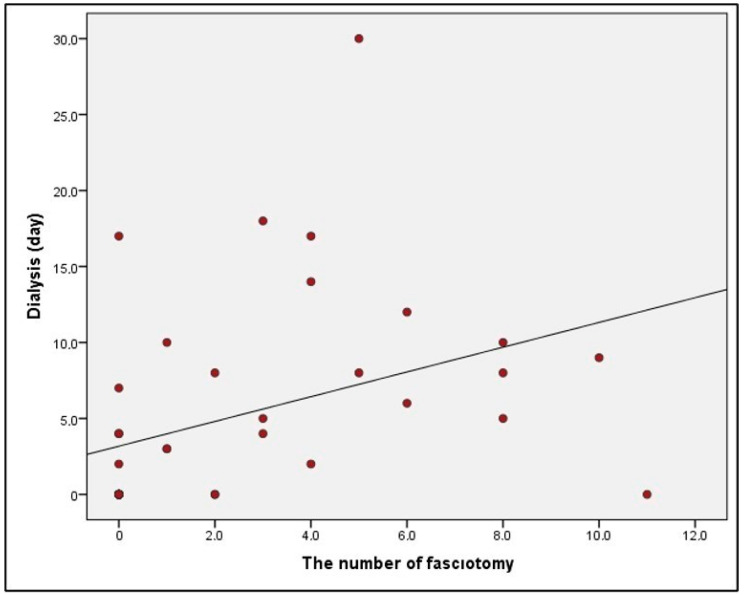
Figure demonstrating the correlation between the number of fasciotomies performed and the duration of dialysis treatment.

**Figure 2 medicina-59-01593-f002:**
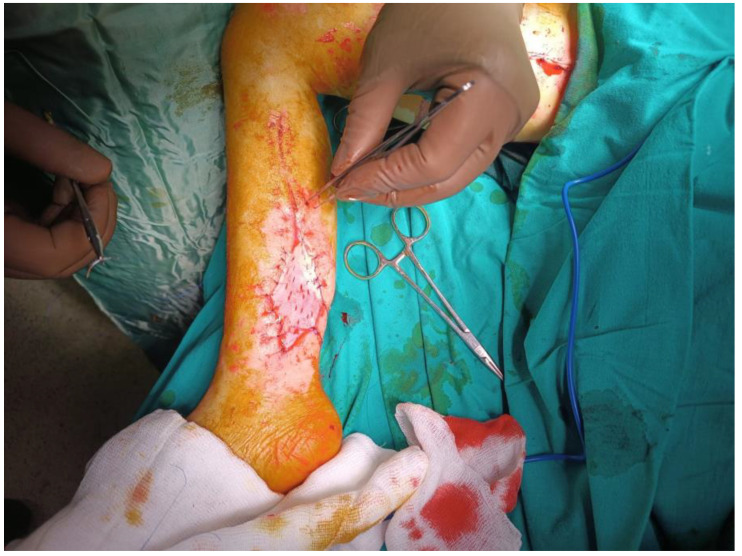
Image presenting the postoperative appearance of male patient, aged four years, after the procedure of skin grafting.

**Table 1 medicina-59-01593-t001:** The distribution of general characteristics of the patients. DIC: Disseminated intravascular coagulation. ARDS: Acute respiratory distress syndrome.

		Min–Max	Mean ± SD (Median)
Age		4–83	32.63 ± 18.99
		N	%
Gender	Male	21	52.5
	Female	19	47.5
Time-to-admission from the earthquake (TAE) (hours)		14–144	44.5 ± 25.87 (36)
Time under the debris (TUD) (hours)		2–40	11.68 ± 8.29 (9.5)
The number of fasciotomy		0–11	2.43 ± 3.16 (1)
Fasciotomy	Absent	19	47.5
	Present	21	52.5
Death	Absent	35	87.5
	Present	5	12.5
The reason for death (n = 5)	Cardiac arrest	3	60
	DIC and Cardiac Arrest	1	20
	Sepsis and DIC	1	20
Nerve Injury	Peroneal nerve injury	23	57.5
	Ulnar nerve injury	8	20
	Radial nerve injury	8	20
	Medain nerve injury	7	17.5
Dialysis day		0–30	5.15 ± 6.72 (3)
DIC	Absent	33	82.5
	Present	7	17.5
ARDS	Absent	32	80
	Present	8	20
Sepsis	Absent	32	80
	Present	8	20
Amputation	Absent	38	95
	Present	2	5

**Table 2 medicina-59-01593-t002:** The comparison of children and adults regarding general characteristics.

		Children	Adults	*P*
		(Min–Max)–(Mean ± SD (Median))	(Min–Max)–(Mean ± SD (Median))	
Age		(4–17)–(10.6 ± 4.48)	(19–83)–(39.97 ± 15.98)	^1^ 0.000 *
		n (%)	n (%)	
Gender	Male	6 (60%)	15 (50%)	^2^ 0.429
	Female	4 (40%)	15 (50%)	
Time-to-admission from the earthquake (TAE) (hours)		(18–144)–(65.1 ± 36.18 (60))	(14–70)–(37.63 ± 17.29 (32))	^3^ 0.011 *
Time under the debris(TUD) (hours)		(8–40)–(16.9 ± 12.16 (10))	(2–30)–(9.93 ± 5.82 (8.5))	^3^ 0.067
The number of fascıotomy		(0–11)–(3.9 ± 4.33 (2))	(0–10)–(1.93 ± 2.57 (0.5))	^3^ 0.261
		n (%)	n (%)	
Fasciotomy	Absent	4 (40%)	15 (50%)	^2^ 0.429
	Present	6 (60%)	15 (50%)	
Death	Absent	10 (100%)	25 (83.3%)	-
	Present	0 (0%)	5 (16.7%)	
The reason for death	Cardiac arrest	-	3 (60%)	-
	DIC and Cardiac Arrest	-	1 (20%)	
	Sepsis and DIC	-	1 (20%)	
Nerve Injury	Peroneal nerve injury	6 (60%)	17 (56.7%)	^2^ 1.000
	Ulnar nerve injury	1 (10%)	7 (23.3%)	-
	Radial nerve injury	0 (0%)	8 (27.7%)	-
	Medain nerve injury	0 (0%)	7 (23.3%)	-

^1^ Student *t* Test, ^2^ Fisher’s Exact Test, ^3^ Mann–Whitney U Test * *p* < 0.05.

**Table 3 medicina-59-01593-t003:** The comparison of paediatric and adult morbidity and serum parameters.

		Children	Adults	*P*
		(Min–Max)–(Mean ± SD (Median))	(Min–Max)–(Mean ± SD (Median))	
Dialysis day		(0–10)–(2.3 ± 3.89 (0))	(0–30)–(6.1 ± 7.24 (4))	^1^ 0.085
		n (%)	n (%)	
DIC	Absent	8 (80%)	25 (83.3%)	^2^ 0.572
	Present	2 (20%)	5 (16.7%)	
ARDS	Absent	10 (100%)	22 (73.3%)	-
	Present	0 (0%)	8 (26.7%)	
Sepsis	Absent	9 (90%)	23 (76.7%)	-
	Present	1 (10%)	7 (23.3%)	
Amputation	Absent	10 (100%)	28 (93.3%)	-
	Present	0 (0%)	2 (6.7%)	
Serum CPK *_(median)_*		(18,790–78,090)–(34,268.1 ± 18,756.8 (28,172))	(2609–56,421)–(23,756.9 ± 12,105.17 (22,943.5))	^1^ 0.092
Serum LDH *_(median)_*		(438–6986)–(2528.9 ± 2096.58 (1843))	(311–8427)–(1990.3 ± 1950.03 (1203))	^1^ 0.399
Serum AST *_(median)_*		(74–4415)–(1833.6 ± 1823.47 (943.5))	(27–2618)–(651.63 ± 634.68 (553.5))	^1^ 0.070
Serum BUN *_(median)_*		(18–122)–(59.9 ± 41.49 (43.5))	(9–221)–(74.23 ± 44.47 (66.5))	^1^ 0.295
Serum Creatinine *_(median)_*		(0.2–2.3)–(1.04 ± 0.82 (0.6))	(0.27–5.36)–(1.9 ± 1.49 (1.4))	^1^ 0.055
Serum Calsium		(5.7–9.9)–(7.6 ± 1.2)	(5.8–9.8)–(7.62 ± 1.01)	^3^ 0.966
Serum Phosphor *_(median)_*		(3.18–9.09)–(5.49 ± 2.28 (4.4))	(2.18–13.45)–(5.32 ± 2.62 (5))	^1^ 0.851
Serum Sodium		(124–140)–(132.2 ± 4.16)	(128–151)–(137.17 ± 5.47)	^3^ 0.012 *
Serum Uric Acid		(4.59–14.51)–(8.42 ± 3.73)	(1.51–13.32)–(7.76 ± 3.21)	^3^ 0.593

^1^ Mann–Whitney U Test, ^2^ Fisher’s Exact Test. ^3^ Student *t* Test. CPK: Creatine phosphokinase. LDH: lactate dehydrogenase. AST: Aspartate aminotransferase. BUN: Blood urea nitrogen. * *p* < 0.05.

**Table 4 medicina-59-01593-t004:** Analysing mortality and morbidity in children and adults based on fasciotomy status.

		Fasciotomy (Children)		*P*
		Absent	Present	
		(Min–Max)–(Mean ± SD (Median))	(Min–Max)–(Mean ± SD (Median))	
Dialysis day		(0–0)–(0 ± 0 (0))	(0–10)–(3.83 ± 4.49 (2.5))	-
		n (%)	n (%)	
DIC	Absent	4 (100%)	4 (66.7%)	^2^ 0.467
	Present	0 (0%)	2 (33.3%)	
ARDS	Absent	4 (100%)	6 (100%)	-
	Present	-	-	
Sepsis	Absent	4 (100%)	5 (83.3%)	^2^ 0.600
	Present	0 (0%)	1 (16.7%)	
Amputation	Absent	4 (100%)	6 (100%)	-
	Present	-	-	
Death	Absent	4 (100%)	6 (100%)	-
	Present	-	-	
		Fasciotomy (Adults)		*P*
		Absent	Present	
		(Min–Max)–(Mean ± SD (Median))	(Min–Max)–(Mean ± SD (Median))	
Dialysis day		(0–17)–(2.27 ± 4.61 (0))	(2–30)–(9.93 ± 7.47 (8))	^1^ 0.000 *
		n (%)	n (%)	
DIC	Absent	13 (86.7%)	12 (80%)	^2^ 1.000
	Present	2 (13.3%)	3 (20%)	
ARDS	Absent	13 (86.7%)	9 (60%)	^2^ 0.215
	Present	2 (13.3%)	6 (40%)	
Sepsis	Absent	13 (86.7%)	10 (66.7%)	^2^ 0.195
	Present	2 (13.3%)	5 (33.3%)	
Amputation	Absent	14 (93.3%)	14 (93.3%)	-
	Present	1 (6.7%)	1 (6.7%)	
Death	Absent	13 (86.7%)	12 (80%)	^2^ 0.500
	Present	2 (13.3%)	3 (20%)	

^1^ Mann–Whitney U Test. ^2^ Fisher’s Exact Test. * *p* < 0.05.

**Table 5 medicina-59-01593-t005:** Assessment of morbidity parameters in relation to fasciotomy number.

		The Number of Fasciotomy	
		Min–Max	Mean–SD (Median)
DIC	Absent	0–11	2.03–2.8 (0)
	Present	0–10	4.29–4.27 (3)
	*p*		0.157
ARDS	Absent	0–11	2.22–3.15 (0)
	Present	0–10	3.25–3.28 (3)
	*p*		0.244
Sepsis	Absent	0–8	1.78–2.59 (0)
	Present	0–11	5–4.07 (5)
	*p*		0.028 *
Amputation	Absent	0–11	2.53–3.21 (1)
	Present	0–1	0.5–0.71 (0.5)
	*p*		0.511

Mann–Whitney U Test. * *p* < 0.05.

## Data Availability

Not applicable.

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
