# Peer review of "A Comparative Analysis of Fasciotomy Results in Children and Adults Affected by Crush-Induced Acute Kidney Injury following the Kahramanmaraş Earthquakes"

_medicina, 2023, doi:10.3390/medicina59091593_

Round 1
Reviewer 1 Report
This is an interesting and well written manuscript. However, the authors have not provided the definitions they used for DIC, ARDS and Sepsis. These should be provided.
The tables are difficult to read and should be reformatted with each table on a separate page.
Author Response
Responses to the Reviewer 1
Thanks for the comments. The responses to your comments are as follows:
- The definitions of DIC, ARDS, and sepsis, as well as the collaborative departments within the hospital involved in establishing these definitions, are clearly indicated in bold between lines 134-165. The definitions were accompanied by references, and the list of references was subsequently reorganised.
- The tables have been rearranged to ensure that each page contains only one table.

Reviewer 2 Report
This retrospective study analyzed the impact of fasciotomy on mortality and morbidity in 40 patients (21 males, 19 females) with crush-induced acute kidney injury (AKI) following the 2023 KahramanmaraÅŸ earthquakes in Turkey. The patients were divided into pediatric (under 18 years) and adult (over 18 years) groups. 21 patients underwent fasciotomy, with 0-11 incisions per patient. The mortality rate was 12.5% (5 adults), with no deaths among pediatric patients. Fasciotomy did not increase mortality in either group. In adults, fasciotomy correlated with longer dialysis duration. The number of fasciotomy incisions significantly correlated with sepsis risk. Despite delayed hospital admission, pediatric patients had comparable mortality and morbidity to adults. In conclusion, fasciotomy did not increase mortality in crush-induced AKI patients after earthquakes. More fasciotomy incisions increased sepsis risk. Pediatric patients had relatively low morbidity and mortality despite admission delays.
Potential limitations of this study and ways it could be improved:
- Small sample size (only 40 patients) from a single center makes it difficult to draw broad conclusions. A larger multi-center study would provide more robust results.
- Retrospective design relies on medical records rather than systematic data collection. A prospective study collecting standardized data would be more rigorous.
- No standard criteria used to define acute kidney injury. Using established criteria like RIFLE or AKIN would improve reliability.
- Potential variability between surgeons in decision to perform fasciotomy and follow-up care. Standardizing surgical indications and management could reduce confounding.
- Limited follow-up time. Longer-term assessment of outcomes like renal function, infections, or limb viability would be informative.
- No patient-reported functional outcomes assessed. Evaluating quality of life, pain, disability would give a fuller picture.
- No multivariate analysis controlling potential confounders like injury severity, comorbidities, etc. This could clarify the independent impact of fasciotomy.
- Unclear if all relevant complications were captured. Prospective study with active surveillance could provide more complete adverse event data.
- Small pediatric subgroup limits conclusions about children. Larger pediatric sample or age-stratified analysis would be helpful.
- No cost-effectiveness analysis. Examining healthcare resource utilization could inform surgical decision-making and disaster planning.
Author Response
Responses to the Reviewer 2
Thanks for the comments. The responses to your comments are as follows:
- The small sample group, which is one of the important limitations of the current study, is expressed in bold on line 373 in the limitation tab of the study. The majority of the patients in the current study were from the neighbouring provinces and the city where the hospital is located was minimally affected by the earthquake. The need for prospective studies with a larger number of patients is stated in bold between lines 395-397 in the conclusion section of the study.
- The challenges associated with performing prospective research in chaotic scenarios, such as earthquakes, are well acknowledged. Furthermore, it is very difficult to obtain approval from ethics committees for prospective research in the context of significant catastrophes, such as earthquakes, within the Turkish setting. The retrospective nature of this research is acknowledged in the limitations section, while the conclusion section emphasises the need for prospective investigations.
- The criteria employed in our study were based on the research conducted by Sever et al. following the 1999 Marmara earthquake. We have appropriately referenced their work in our study. There are also articles available that discuss the applicability of RIFLE criteria in challenging scenarios, such as earthquake situations. One of the limitations of our study is the absence of standardised criteria, which is clearly stated in the limitations section.
- The determination regarding the implementation of fasciotomy was made based on in-clinic evaluations conducted by physicians having a minimum of five years of experience in the field of trauma medicine. The process was managed by a group of experienced physicians within the clinic, rather than being solely reliant on the decision of an individual surgeon. The aforementioned information is highlighted in bold within lines 100-102.
- The study's limited follow-up period is clearly stated in bold within the limitations section, specifically in line 373. It would be highly advantageous for us to undertake a comprehensive study that examines the long-term effects. However, the majority of patients expressed their voluntary desire to seek treatment in other cities once they had successfully overcome the acute phase. It should be noted that no patient was able to be monitored for a duration exceeding four months. Due to the constraints of this situation, we were restricted in our ability to undertake a comprehensive study examining the long-term effects.
- The assessment of patient-reported functional outcomes was not conducted. The majority of our patients were receiving psychiatric assistance. The patient profile comprised individuals who have experienced the loss of their children while still being alive, as well as children who have lost their parents. Given the potential impact of these psychological effects on patient-reported functional outcomes, we did not allocate any resources towards addressing this matter.
- The research eliminated individuals who had a documented medical history of illness or comorbidity, as shown in bold text between lines 91-92. The calculation of Injury Severity scores was not feasible during the management of the earthquake, given the inherently chaotic nature of the process.
- In light of the retrospective nature of the study, an analysis was conducted on hospital records, consultation notes from relevant departments, and the orthopaedic surgery clinic's own notes. Additionally, the daily follow-up notes of intensive care doctors were examined during the intensive care follow-up. Based on our analysis, it is our belief that the observed decrease in the incidence of complications is not of significant concern.
- A significant constraint of the research lies in the limited sample size of the participants. It is well acknowledged that a considerable number of individuals afflicted with injuries resulting from earthquakes seek medical attention at various healthcare facilities. Multicentre studies may potentially provide more value to the existing body of knowledge. Nevertheless, it is unfeasible to expand the patient population under the current circumstances.
- One limitation of our study is the lack of cost-effectiveness analysis, as indicated in bold between lines 382-385 in the limitation section.

Round 2
Reviewer 1 Report
The manuscript is acceptable.
Reviewer 2 Report
It appears that all comments have been appropriately responded to. I have no further comments and recommend publication.